# Exhaled Breath Analysis Detects the Clearance of *Staphylococcus aureus* from the Airways of Children with Cystic Fibrosis

**DOI:** 10.3390/biomedicines12020431

**Published:** 2024-02-14

**Authors:** Elias Seidl, Johann-Christoph Licht, Rianne de Vries, Felix Ratjen, Hartmut Grasemann

**Affiliations:** 1Division of Respiratory Medicine, Department of Pediatrics, The Hospital for Sick Children, Toronto, ON M5G 1X8, Canada; eliasseidl@hotmail.de (E.S.); johann.licht@mail.utoronto.ca (J.-C.L.); felix.ratjen@sickkids.ca (F.R.); 2Division of Respiratory Medicine, University Children’s Hospital Zurich, 8032 Zurich, Switzerland; 3Breathomix BV, Bargelaan 200, 2333 CW Leiden, The Netherlands; rianne.devries@breathomix.com; 4Translational Medicine Program, Research Institute, The Hospital for Sick Children, University of Toronto, Toronto, ON M5G 1X8, Canada

**Keywords:** electronic nose, cystic fibrosis, respiratory disease, respiratory infections, volatile organic compounds, *Staphylococcus aureus*

## Abstract

Background: Electronic nose (eNose) technology can be used to characterize volatile organic compound (VOC) mixes in breath. While previous reports have shown that eNose can detect lung infections with pathogens such as *Staphylococcus aureus* (SA) in people with cystic fibrosis (CF), the clinical utility of eNose for longitudinally monitoring SA infection status is unknown. Methods: In this longitudinal study, a cloud-connected eNose, the SpiroNose, was used for the breath profile analysis of children with CF at two stable visits and compared based on changes in SA infection status between visits. Data analysis involved advanced sensor signal processing, ambient correction, and statistics based on the comparison of breath profiles between baseline and follow-up visits. Results: Seventy-two children with CF, with a mean (IQR) age of 13.8 (9.8–16.4) years, were studied. In those with SA-positive airway cultures at baseline but SA-negative cultures at follow-up (*n* = 19), significant signal differences were detected between Baseline and Follow-up at three distinct eNose sensors, i.e., S4 (*p* = 0.047), S6 (*p* = 0.014), and S7 (*p* = 0.014). Sensor signal changes with the clearance of SA from airways were unrelated to antibiotic treatment. No changes in sensor signals were seen in patients with unchanged infection status between visits. Conclusions: Our results demonstrate the potential applicability of the eNose as a non-invasive clinical tool to longitudinally monitor pulmonary SA infection status in children with CF.

## 1. Introduction

Cystic fibrosis (CF) is a genetic disease of autosomal recessive inheritance that affects multiple organ systems including the sweat glands, pancreas, liver and intestines, reproductive tract, sinuses, and lungs [1,2,3]. CF is caused by pathogenic variants in the cystic fibrosis transmembrane conductance regulator (*CFTR*) gene located on the long arm of human chromosome 7 (7q31.2). The CFTR protein, a member of the adenosine-triphosphate (ATP)-binding cassette (ABC) transporter superfamily, is a cAMP-regulated epithelial anion channel composed of 1480 amino acids and is expressed in secretory cells and exocrine glands in the human body [4]. In the airways, CFTR facilitates the transport of chloride ions and bicarbonates through epithelial cell membranes and regulates the activity of the epithelial sodium channel (ENaC) [5,6,7]. CFTR dysfunction in the lung leads to an imbalance in water–electrolyte homeostasis and to airway surface dehydration. CF lung disease is characterized by impaired mucociliary clearance, airway inflammation, and infection with opportunistic pathogens, and remains the main cause of morbidity and mortality in affected individuals [8,9,10]. 

Infections of the CF lung with *Staphylococcus aureus* (SA), a Gram-positive bacterium, often occur early in the disease process, and SA is the most common pathogen identified in the respiratory tract of children with CF [11]. While the potential benefits of prophylactic strategies remain controversial [12,13,14], evidence suggests that SA infection can lead to significant damage and lung function decline in people with CF (pwCF) [11]. 

Another important opportunistic pathogen causing lung infections in CF is *Pseudomonas aeruginosa* (PA), which is known to contribute to CF lung disease progression [15,16]. PA produces and consumes specific VOCs [17,18], and PA airway infection in CF is associated with a higher pro-inflammatory response compared to other pathogens [19,20]. Previous studies have shown that eNose technology is also able to detect PA infection in pwCF with a high accuracy (ROC/AUC of 0.69 to 0.93) [20,21].

The recent introduction of CFTR modulator therapy has resulted in remarkable improvements in clinical endpoints, including nutritional status and respiratory outcomes, in treated pwCF [22,23]. However, effective modulator therapy frequently results in inability in treated individuals to provide an expectorated sputum sample at the clinical point of care (POC), which affects routine microbiology testing particularly in children with CF and those with early or mild disease manifestation.

Human-exhaled breath contains a variety of volatile organic compounds (VOCs), including alkanes, benzene derivatives, acetone, dimethyl sulphide, phenol, and aromatic compounds [24]. The individual composition of VOCs in breath has been found to be altered in numerous medical conditions, with or without lung infection, as was recently reviewed [25]. VOC analysis in breath using electronic nose (eNose) technology represents a non-invasive tool potentially applicable to patients at the POC. In the context of CF, studies have found that VOC breath profiles from pwCF can be distinguished from those of healthy controls [26] and from patients with other lung diseases, such as asthma [27] or primary ciliary dyskinesia [28]. In a recent observational study, we demonstrated that eNose can differentiate children with CF with an airway culture positive for SA from those with negative cultures [29]. To further define the potential clinical utility of this technology in monitoring infection status in children with CF over time, the aim of this study was to investigate whether the clearance of SA from the airways of patients with a previous airway culture positive for SA can also be detected using the eNose.

## 2. Materials and Methods

### 2.1. Design and Study Population

This was a prospective longitudinal study in children with CF followed at the outpatient clinic of the Hospital for Sick Children (SickKids) in Toronto, Canada, between February 2020 and November 2022. The study was approved by the SickKids Research Ethics Board (REB #1000064324). All clinically stable CF patients between the age of 5 and 18 years for whom consent had been given and who were able to perform the eNose respiratory maneuvers were eligible for this study. Patients attending routine CF clinic were approached and asked to participate. There were no restrictions in terms of eating or drinking in the hours before the measurement. The diagnosis of CF was made in accordance with current guidelines [30,31] and was confirmed with sweat chloride testing via quantitative pilocarpine iontophoresis (>60 mEq/L), and/or *CFTR* genotyping with two identifiable CF mutations. Patients with a physician diagnosis of a CF pulmonary exacerbation at either study visit were excluded. Patients were instructed to perform two measurement maneuvers into a SpiroNose device, as described previously [29], following pulmonary function testing (PFT) via spirometry as part of clinical routine care. PFTs were performed following ATS guidelines [32], and forced expiratory volume in one second (FEV_1_) was calculated and expressed as percent predicted (ppFEV_1_) using the GLI equations [33]. Demographic data and clinical characteristics were obtained through medical chart review. Spontaneously expectorated sputum samples or throat swabs were collected at each visit and processed for microbiology testing as per routine clinical care. Microbiology culture results were labeled as positive for SA, *Pseudomonas aeruginosa* (PA), *Haemophilus influenzae*, *Acinetobacter species*, *Burkholderia cepacia* complex, *Mycobacterium abscessus*, *Aspergillus fumigatus*, or no CF pathogens (no growth or usual respiratory flora after 3 days of culture). Airway cultures were obtained and processed following recommendations of the Microbiology Consensus Conference of the Cystic Fibrosis Foundation [34]. The specimens were inoculated onto agar plates via swab and streaked for isolation of colonies. Media inoculated were 5% sheep blood agar, chocolate agar, colistin/nalidixic acid agar, MacConkey agar, mannitol salt agar, and oxidation–fermentation polymyxin bacitracin lactose agar (all from Remel, Lenexa Kansas). Agar plates were incubated at 35 degrees centigrade and observed for growth of pathogens for a minimum of 72 h. 

### 2.2. Exhaled Breath Analysis

Breath profiles were analyzed using the BreathBase System (Breathomix, Leiden, The Netherlands), a combination of an eNose (SpiroNose, Breathomix, Leiden, The Netherlands) and an online BreathBase Platform [29,35]. The technical setup for the SpiroNose has been described in detail before [35]. The eNose consists of 7 different cross-reactive metal oxide semiconductor (MOS) sensors (Figaro, Osaka, Japan). These 7 sensors are each present in duplicate on the inside (to measure VOCs in exhaled breath) and on the outside of the eNose (to measure VOCs in ambient air). Using quality control gas, both the operation of the eNose sensors and the absence of sensor drift were verified monthly.

### 2.3. eNose Respiratory Maneuvers 

The eNose measurement setup used in this study included a nose clamp and a mouthpiece with a viral/bacterial filter (Pulmosafe V3/2, Lemon Medical GmbH, Hammelburg, Germany) attached to the SpiroNose (Breathomix, Leiden, The Netherlands). Patients were asked to rinse their mouth thoroughly with water three times before exhaling into the eNose. Participants were instructed to, after putting on a nose clip, perform five tidal breaths followed by deep inspiration to total lung capacity, a 5 s breath hold, slow (<0.4 L·s^−1^) maximal expiration towards residual volume and a final deep inhalation into the device via a bacterium and virus filter twice with a 2 min interval between repeat maneuvers. Exhaled breath profiles were measured in real time via the SpiroNose and the obtained sensor signal data were sent and stored directly in the online BreathBase Platform [35]. 

### 2.4. Data Processing

Processing of the eNose sensor signals is performed automatically via the standard eNose software (www.breathomix.com) as previously published [35], and includes filtering of the sensor signals in both time and frequency domains, de-trending and alignment of the sensor signals, and advanced environmental VOC correction. From each sensor signal, two variables are determined: (1) the highest sensor peak normalized to the most stable sensor, sensor 2, to minimize inter-array differences, and (2) the ratio between sensor peak and low point obtained during breath hold (BH). Both variables have discriminative power. The sensor peak and peak/BH ratios are included in a .csv file that serves as the source document for statistical analysis [35]. 

### 2.5. Statistical Analysis

Statistical analysis was performed in SPSS (IBM Corp. Released 2020. IBM SPSS Statistics for Windows, Version 28.0. Armonk, NY, USA: IBM Corp.). For the longitudinal analysis, patients were allocated into three distinct groups based on airway microbiology results: Group 1 (no SA at Baseline and Follow-up), Group 2 (SA positivity at Baseline and Follow-up), and Group 3 (SA positivity at Baseline but no SA at Follow-up). 

Descriptive statistics were expressed as number (percent) or median (IQR). Group differences were tested with the Kruskal–Wallis test or Chi-square test, as appropriate. The normalized sensor peaks and peak/BH ratios were compared between visits (Baseline and Follow-up) by means of independent sample t-tests internally validated using 1000 iterations of bootstrap. Linear discriminant analysis with leave-one-out cross-validation using the significant sensor peaks and peak/BH ratios was carried out. Cross-validation was used to assess the linear discriminant model. The discriminant scores were used for receiver operating characteristic (ROC) analysis and to determine the area under the curves (AUC-ROC).

## 3. Results

A total of 72 (35 males; 48.6%) children were included. The baseline characteristics for all study participants are shown in Table 1. At Baseline, 53 (73.6%) patients and at Follow-up, 47 (65.3%) participants were able to expectorate sputum for airway microbiology testing spontaneously. Throat swabs were collected from the remaining 19 (26.4%) children, as per clinical routine. At Baseline, 8 children had negative culture results for CF pathogens (sterile cultures or usual respiratory flora) from microbiology testing; 64 were positive for any CF pathogen (50 mono-infections, 14 co-infections). Mono-infections included SA (*n* = 47), *Pseudomonas aeruginosa* (PA) (*n* = 1), and *Aspergillus* species (*n* = 2); co-infections of other pathogens with SA included PA (n = 7), *Haemophilus influenzae* (*n* = 2), *Acinetobacter* species (*n* = 1), *Achromobacter* species (*n* = 1), *Mycobacterium abscessus* (*n* = 1), *Aspergillus fumigatus* (*n* = 1), and SA with PA and *Aspergillus fumigatus* (*n* = 1). There was no significant difference between groups in terms of demographics or treatments (Table 1). A CONSORT diagram of the study is shown in Figure 1.

At Baseline, the eNose was able to distinguish between children with CF who were culture-positive for SA (*n* = 61) and -negative for SA (*n* = 11) in sensors S7 (*p* < 0.001) and S7BH (*p* = 0.008) (AUC 0.757, CI95% 0.617–0.898). The longitudinal analysis revealed similar breath profiles at Baseline and Follow-up visits with no significant difference for any sensor for Group 1 (no SA at Baseline and Follow-up) and Group 2 (SA positivity at Baseline and Follow-up). In contrast, in Group 3 (SA positivity at Baseline, no SA at Follow-up), significant signal differences were detected between Baseline and Follow-up for three distinct eNose sensors, i.e., S4 (*p* = 0.047), S6 (*p* = 0.014), and S7 (*p* = 0.014) (Figure 2). In Group 3, six patients received antibiotic treatments between visits: nebulized inhaled tobramycin in three, as well as oral azithromycin and caphadroxil in one patient, respectively. The breath profiles at Follow-up did not differ between children who received and those who did not receive antibiotics between visits.

## 4. Discussion

This is, to our knowledge, the first longitudinal study evaluating the potential of an eNose to monitor children with CF for changes in their respiratory infection status. We not only confirm that the SpiroNose can differentiate between children with CF who are airway-culture-positive or -negative for *Staphylococcus aureus* (SA), but we also demonstrate, for the first time, that the clearance of SA from CF airways resulted in a detectable change in VOC composition in breath. Our study, therefore, adds to previous work by suggesting that eNose technology can potentially be used to monitor infection status in pwCF. 

The concept of using breath analyses to detect lung infection and the clearance of infection non-invasively is appealing. Currently, the routine monitoring of infection status in pwCF relies on the microbiology testing of expectorated sputum or throat/cough swabs in those unable to expectorate. However, obtaining lower-airway samples can be challenging, particularly in younger children. In our study population, over one quarter of the participants were unable to expectorate sputum spontaneously at the clinical POC, a number that is likely to further increase due to the recent approval of effective CFTR modulator therapy that results in a significant reduction in sputum production in treated pwCF [36,37,38]. In this study, we were able to show that repeat breath profiles at follow-up remained unchanged in children with CF with stable airway infection status, i.e., those with either SA-positive or SA-negative cultures at both study visits, while the clearance of SA in patients previously positive for SA was associated with a significant change in breath profile. 

The SpiroNose contains a total of 14 sensors able to detect VOCs. In this study, signal differences in sensors S7 and S7BH (AUC 0.757, CI95% 0.617–0.898) were associated with SA positivity at Baseline. In a recent study from our group using the SpiroNose, SA positivity was also associated with signal alterations in sensors S7 and S4. In contrast, infection with *Pseudomonas aeruginosa* (PA), another common opportunistic CF pathogen known to contribute to lung disease progression in pwCF [15], was associated with differences in sensor S5 [29]. In the current study, the clearance of SA at Follow-up resulted in signal changes in sensors S4, S6, and S7. The breath profiles at Follow-up did not differ between children who had cleared SA infection following antibiotic therapy and those who cleared the infection spontaneously. However, the antibiotic treatments used were not targeting SA specifically, and the number of patients in these sub-groups was low. Together, these findings are in line with previous studies reporting that CF pathogens including SA, PA, and *Aspergillus fumigatus* can be detected and differentiated using eNoses with high accuracy [20,21,26,27,39,40] and suggest that the clearance of SA infection can also be detected. Further studies are needed to analyse whether eNose technology can be used to monitor responses to antibiotic therapy targeting CF airway pathogens with the aim of eradication. 

The current standard for the detection of pathogens in routine CF clinical care relies on aerobic microbiology cultures of airway secretions that require a minimum of 72 h of incubation [34]. A potential advantage of eNose technology is that it can be used to monitor airway infection status at the clinical point of care during routine clinic visits. Furthermore, the positive predictive value for lower-airway infection using throat swabs is lower compared to sputum samples and bronchial–alveolar lavage fluid [41], the latter being an expensive, time-consuming, and invasive procedure requiring general anaesthetic. Therefore, eNose technology may provide opportunities to detect lung infections in pwCF unable to expectorate sputum or those with decreased sputum production following treatment with highly effective CFTR modulators. Additional advantages of the eNose compared to current practice also include lower costs, ease of use, and portability [42].

In contrast to gas chromatography mass spectrometry (GC-MS), eNose technology does not identify individual molecular compounds but uses multiple cross-reactive sensors that are not VOC-specific [35]. This technology produces characteristic breath profiles and uses pattern recognition to identify pulmonary diseases with and without infections [25,42,43]. In this study, three distinct sensors were associated with the clearance of SA infection. Sensor S4 (TGS 2600 sensor) has the highest sensitivity to air contaminants including hydrogen, carbon, monoxide, and ethanol; sensor S6 (TGS 2620 sensor) is mainly sensitive to alcohol and solvent vapors; and sensor S7 (TGS 2612 sensor) is mainly sensitive to methane, propane, and iso-butane [35,44]. While eNose technology is a fast and easy-to-use screening tool, the identification of the exact VOCs responsible for changes in breath profiles is of particular interest, as this may lead to a better understanding of the underlying pathophysiology of the disease [45]. 

An earlier study performed the technical validation of the eNose used in this study (the SpiroNose) [46]. The authors found expiratory flow rates between 0.19 and 0.38 L·s^−1^ suitable for measurements, and that there was no need to correct for lung volumes. This was in general agreement with earlier findings [47,48]. To ensure the correct expiratory flow, a visual feedback system is implemented and checked twice, during testing and before data analysis. The development of devices aiming to improve the applicability of the eNose technology to younger age groups unable to perform the currently desired breath maneuver is the focus of current research.

There are several limitations to our study to consider when interpreting the results. The patients participating in this study were recruited from one centre, leading to a relatively small number of participants, which limited the statistical power of some subgroup analyses. However, the number of subjects included here was similar to that in previous publications [21,26,28,29]. Also, we did not analyse breath profiles from patients who acquired SA infections during the observational period as this group was too small to be included. Furthermore, throat swabs to detect lower-airway infection in children unable to produce sputum, while used in clinical routine, may be less sensitive than the gold standard for the microbial sampling of the airways, i.e., bronchoscopy with bronchoalveolar lavage [49], and some SA-positive patients may therefore have been missed and assigned to the SA-negative group for analysis. 

While it is conceivable that changes in eNose signals are not only reflective of infection status, i.e., the persistence of infection or the clearance of pathogens, but also of inflammatory changes secondary to the infection, little is known about the potential contribution of inflammation to breath profiles, even though previous work had shown that VOC mixes can be different between CF patients with sterile airway cultures and healthy controls [29,42,46,50,51]. Future studies are needed to help us understand the contribution of CF airway inflammation to breath profile alterations and to further define the clinical utility of this technology.

In conclusion, our findings suggest that eNose technology may prove to be a useful and non-invasive tool to monitor infection in pwCF during routine clinical testing at the POC. Further studies are needed to provide a better understanding of the exact VOCs responsible for the observed differences and insights into the role of SA in CF lung disease.

## Figures and Tables

**Figure 1 biomedicines-12-00431-f001:**
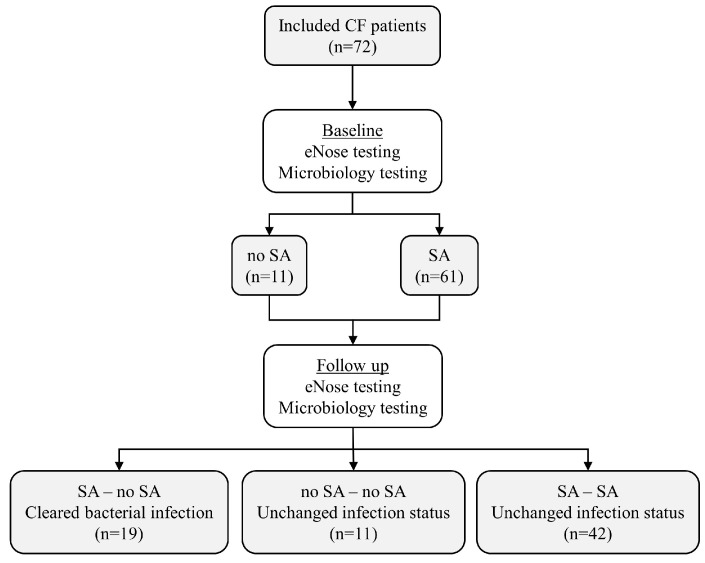
CONSORT diagram of the study. SA: *Staphylococcus aureus*.

**Figure 2 biomedicines-12-00431-f002:**
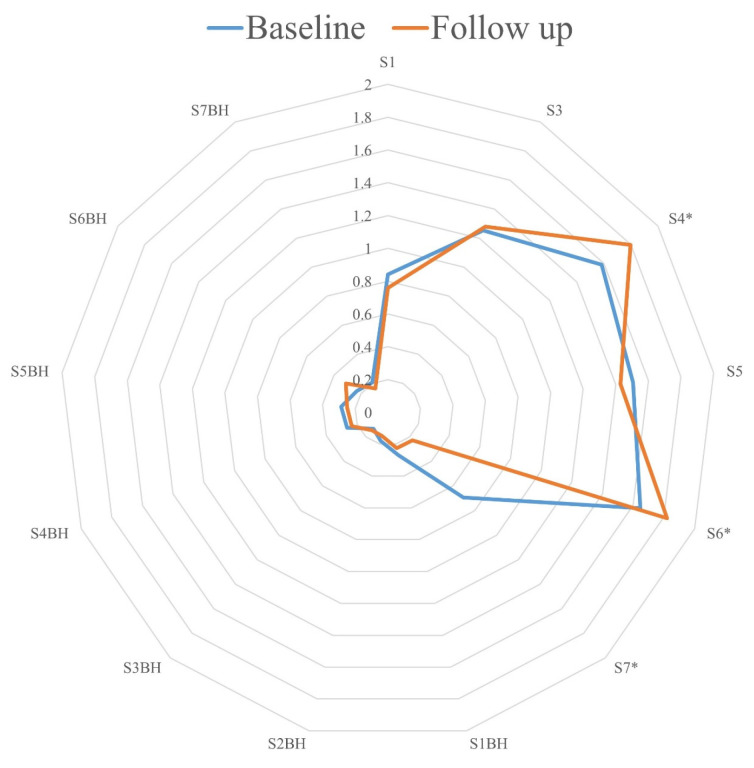
Radar chart visualizing means of the normalized SpiroNose sensors at the Baseline (blue) and Follow-up (orange) visits. The values along each axis of the radar chart are connected linearly to visualize the dataset as a polygon. The asterisk indicates a significant difference between both visits.

**Table 1 biomedicines-12-00431-t001:** Demographics and treatments of CF participants. Data are presented as median (IQR) or mean (SD).

Demographics	Total Cohort	Group 1 (No SA–No SA)	Group 2 (SA–SA)	Group 3 (SA–No SA)	*p* Value ^a^
Included subjects	72	11	42	19	-
Male sex	35 (48.6%)	5 (45.5%)	20 (47.6%)	10 (52.6%)	0.912
Age in years	13.8 (9.8–16.4)	14.8 (10.5–16.2)	15.0 (9.9–16.6)	12.4 (9.4–15.6)	0.254
Preschool (< 5 years)	4 (5.6%)	1 (9.1%)	2 (4.8%)	1 (5.3%)	-
School age (6–12 years)	28 (38.9%)	3 (27.3%)	15 (35.7%)	10 (52.6%)	-
Adolescent (≥13 years)	40 (55.6%)	7 (63.6%)	25 (59.5%)	8 (42.1%)	-
BMI in kg/m^2^	18.6 (16.8–20.3)	19.2 (17.0–21.9)	18.5 (16.8–19.8)	18.5 (15.8–20.7)	0.707
ppFEV_1_	93.0 (78.5–102.0)	82.0 (59.0–10.6.0)	92.5 (81.8–102.3)	97.0 (90–102.0)	0.296
Pancreatic insufficiency	64 (88.9%)	9 (81.8%)	36 (85.7%)	19 (100%)	0.186
Time from baseline to follow-up in months	8.0 (4.6–11.5)	11.2 (6.8–19.2)	8.1 (5.4–11.4)	6.0 (2.0–10.2)	0.077
Treatment					
Saline, hypertonic	38 (52.8%)	7 (63.6%)	19 (45.2%)	12 (63.2%)	0.317
Salbutamol	59 (81.9%)	10 (90.9%)	31 (73.8%)	18 (94.7%)	0.101
Dornase alfa	44 (61.1%)	6 (54.5%)	26 (61.9%)	12 (63.2%)	0.885
Inhaled antibiotics	10 (13.9%)	1 (9.1%)	6 (14.3%)	3 (15.8%)	0.872
Azithromycin	8 (11.1%)	3 (27.2%)	4 (9.5%)	1 (5.3%)	0.159
CFTR modulators	27 (37.5%)	5 (45.5%)	15 (28.6%)	7 (36.8%)	0.836
Ivacaftor	4 (5.6%)	- (0%)	3 (7.1%)	1 (5.3%)	0.653
Ivacaftor/tezacaftor/elexacaftor	11 (15.3%)	2 (18.2%)	6 (14.3%)	3 (15.8%)	0.948
Ivacaftor/lumacaftor	8 (11.1%)	1 (9.1%)	5 (11.9%)	2 (10.5%)	0.961
Ivacaftor/tezacaftor	4 (5.6%)	2 (18.2%)	1 (2.4%)	1 (5.3%)	0.125

Abbreviation: FEV_1_: forced expiratory volume in 1 s, CFTR: cystic fibrosis transmembrane conductance regulator. ^a^ Group differences were tested with the Kruskal–Wallis test or Chi-square test, as appropriate.

## Data Availability

Data are available upon reasonable request.

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
