# Peer review of "Exhaled Breath Analysis Detects the Clearance of Staphylococcus aureus from the Airways of Children with Cystic Fibrosis"

_biomedicines, 2024, doi:10.3390/biomedicines12020431_

Round 1

Reviewer 1 Report

Comments and Suggestions for Authors

This manuscript delineates the exhalation analysis of the Staphylococcus aureus clearance in Cystic Fibrosis paediatric patients. This manuscript is well-written and well-organised. The methodology is sound and very interesting. It could be adapted to the point-of-care and save time for definitive diagnosis. The outcomes are acceptable for publication. They contribute to an area lacking existing literature and are likely to interest readers.

Major concerns.

1. This study focuses on exhalation analysis in paediatric patients. Table 1 must be subgrouped by age group (e.g. Preschool, School-age child and Adolescent).

Moreover, a different age in paediatric means a different lung volume that affects the sensitivity of exhalation analysis.

Comments.

1. I suggest adding the CONSORT flow diagram to make it clearer and more informative.

2. Lines 61-70 and Discussion. You can emphasise the advantages of exhalation analysis, which takes a few minutes to get the result, compared to a culture or molecular approach, which takes several hours.

3. You can add Staphylococcus aureus as one of the keywords in this manuscript.

Typos.

1. Lines 154-158. Consider italicising for genus and species names.

2. Lines 92-93 "Burkholderia cepacia complex", the "complex" is not required to be italicised.

Reviewer 2 Report

Comments and Suggestions for Authors

— The present study has a significant influence, but it needs a minor revision.

- The abstract must illustrate the used methods and the most prevalent results (give more hints about methods and results).

—As many studies have proven, Pseudomonas aeruginosa is one of the main causes of serious problems and even death in people with cystic fibrosis.

Considering this fact, we suggest mentioning other pathogens in the introduction, especially P. aeruginosa.

-second, is the eNose useful in diagnosing Pseudomonal infection?

- Discuss more on the alarming results of this research. 
- The authors are advised to illustrate the real impact of their findings.

Reviewer 3 Report

Comments and Suggestions for Authors

This study evaluates the use of a eNose technology to detect SA breath profiles and the corresponding infection caused by this bacterial species. The study is  well and simply designed and is clear and easy to understand. However I have some concerns:

A Table indicating the results of the culture tests for each patient in comparison with the SA VOCs (elimination or not of SA) would be interesting to compare both approaches.

Because the procedure to obtain the samples for VOCs profiles (lines 112-121) seems to be complex and difficult it would need a comparison with the results obtained using the "standard" bacteria culture method. Also because most of the patients provided expectorate or throat swab, procedures that seem to be easier to perform when compare with the eNose maneuvers.

Authors have to clearly highlight the advantage of using eNose technology especially in young/very young patients that could have difficulties in providing the samples.

What about the other CF pathogens? is it possible to obtain their VOC profiles? and what about the accuracy of the breath test evaluation when the patients are colonized by two or more bacterial species? Please comment on these points.

Lines 191-195: for the same reasons these patients would have difficulties in performing the eNose maneuvers

Lines 213-215:....and to investigate whether the somministration of antibiotics could interfere with the production of the VOC profiles.
